# The Role of Antioxidants Supplementation in Clinical Practice: Focus on Cardiovascular Risk Factors

**DOI:** 10.3390/antiox10020146

**Published:** 2021-01-20

**Authors:** Vittoria Cammisotto, Cristina Nocella, Simona Bartimoccia, Valerio Sanguigni, Davide Francomano, Sebastiano Sciarretta, Daniele Pastori, Mariangela Peruzzi, Elena Cavarretta, Alessandra D’Amico, Valentina Castellani, Giacomo Frati, Roberto Carnevale, SMiLe Group

**Affiliations:** 1Department of General Surgery and Surgical Specialty Paride Stefanini, Sapienza University of Rome, 00185 Rome, Italy; 2Department of Clinical Internal, Anesthesiological and Cardiovascular Sciences, Sapienza University of Rome, 00185 Rome, Italy; simona.bartimoccia@uniroma1.it (S.B.); daniele.pastori@uniroma1.it (D.P.); valentina.castellani@uniroma1.it (V.C.); 3Unit of Internal Medicine and Endocrinology, Madonna delle Grazie Hospital, Velletri, 00049 Rome, Italy; valerio.sanguigni@uniroma2.it (V.S.); davide.francomano@uniroma1.it (D.F.); 4Department of Internal Medicine, University of Rome “Tor Vergata”, 00133 Rome, Italy; 5Department of Medical-Surgical Sciences and Biotechnologies, Sapienza University of Rome, 04100 Latina, Italy; sebastiano.sciarretta@uniroma1.it (S.S.); mariangela.peruzzi@uniroma1.it (M.P.); elena.cavarretta@uniroma1.it (E.C.); giacomo.frati@uniroma1.it (G.F.); 6Department of AngioCardioNeurology, IRCCS Neuromed, 86077 Pozzilli, Italy; 7Mediterranea, Cardiocentro, 80122 Napoli, Italy; 8Department of Movement, Human and Health Sciences, University of Rome “Foro Italico”, 00135 Rome, Italy; a.damico@studenti.uniroma4.it; 9Faculty of Medicine and Surgery, Sapienza University of Rome, 04100 Latina, Italy; smilegrouplatina@libero.it

**Keywords:** oxidative stress, cardiovascular disease, antioxidants, risk factors, biomarkers, supplementation

## Abstract

Oxidative stress may be defined as an imbalance between reactive oxygen species (ROS) and the antioxidant system to counteract or detoxify these potentially damaging molecules. This phenomenon is a common feature of many human disorders, such as cardiovascular disease. Many of the risk factors, including smoking, hypertension, hypercholesterolemia, diabetes, and obesity, are associated with an increased risk of developing cardiovascular disease, involving an elevated oxidative stress burden (either due to enhanced ROS production or decreased antioxidant protection). There are many therapeutic options to treat oxidative stress-associated cardiovascular diseases. Numerous studies have focused on the utility of antioxidant supplementation. However, whether antioxidant supplementation has any preventive and/or therapeutic value in cardiovascular pathology is still a matter of debate. In this review, we provide a detailed description of oxidative stress biomarkers in several cardiovascular risk factors. We also discuss the clinical implications of the supplementation with several classes of antioxidants, and their potential role for protecting against cardiovascular risk factors.

## 1. Introduction

Oxidative stress has an important role in the onset and in the progression of several diseases, and in particular, in cardiovascular diseases. Oxidative stress is caused by the overproduction of reactive oxygen species (ROS), which include both the free radicals and their non-radical intermediates, such as superoxide anion (O_2_^•−^), hydroxyl ion (OH^•^), hydrogen peroxide (H_2_O_2_), and peroxyl radicals (ROO^•^), alkoxyl (RO^•^), singlet oxygen (^1^O_2_), and ozone (O_3_). The burst of ROS is associated with an imbalance between the generated ROS and the antioxidant defense systems. Evidence shows that oxidative stress plays an important role in the progression of various cardiovascular diseases, such as atherosclerosis, heart failure (HF), cardiac arrhythmia, and myocardial ischemia-reperfusion (I/R) injury. A lot of work has been devoted to the studies of antioxidants therapies in the prevention and treatment of these cardiovascular diseases. While some clinical trials have shown positive results, others are controversial.

This is partly due to the incorrect evaluation of biomarkers of oxidative stress, and in particular, to the lack of the assessment of key ROS-producing enzymes, such as NADPH oxidase. Furthermore, the choice of the dosage and type of antioxidant used in the treatment or prevention of cardiovascular diseases is not accurate and specific for each pathology.

In this review, we will discuss the main biomarkers of oxidative stress used in clinical practice and their therapeutic implications in cardiovascular diseases. We will also highlight the different antioxidant treatments and their effect on the type of biomarker and cardiovascular risk factors.

## 2. Biomarkers of Oxidative Stress in Clinical Practice

Reactive oxygen species (ROS) are key cellular components that play an important role in various physiological conditions, as well as in the development of several diseases [1]. The ROS play a dual role, both beneficial and toxic to the organism. At moderate or low levels, ROS have beneficial effects and act on various physiological functions like immune function (i.e., defense against pathogenic microorganisms), in some number of intracellular pathways, and in redox regulation. Conversely, high concentrations of ROS induce oxidative stress, a pathological condition characterized by an overload of free radicals that are not neutralized, have a toxic effect, and modify the integrity of cell membranes and other structures, such as organic macromolecules [2]. Oxidative stress is responsible for numerous chronic and degenerative diseases, such as cancer, autoimmune disorders, rheumatoid arthritis, aging, neurodegenerative and cardiovascular diseases. Considering the evidence about the association between oxidative stress with a multitude of human diseases, the measurement of oxidative stress biomarkers plays a pivotal role in the evaluation of the health status, as well as the development of oxidative stress-mediated disorders [3]. Many methods have been developed and used to measure the concentration and nature of biomarkers of oxidative stress (Table 1).

Although many biomarkers can be used to determine oxidative stress, most of these benchmarks are limited in vivo [20,21]. The precise measurement of ROS in vascular cells and tissues represents a challenge because of their low levels and transient lifetimes [22]. Indeed, when produced within living cells, the short half-life (seconds) of a certain ROS limits the distance it can diffuse and thereby its radius of action. This means that the direct reaction of a short-lived ROS like O_2_^•−^ in situ is likely restricted to a small sub-cellular volume surrounding the site of its generation (“local” ROS), whereas ROS with a longer half-life like H_2_O_2_ might be more suited for global signaling [23]. Moreover, the efficient systems scavenging ROS require that any detection method must be sensitive enough and must allow in situ measurements in the tissues of interest.

To evaluate ROS production, we can use several probes, such as (1) dihydrochlorofluorescein diacetate (DCFH-DA), which is used to detect H_2_O_2_, hydroxyl radicals (OH-), and peroxyl radicals (ROO-); (2) dihydrorhodamine 123, which is another uncharged and nonfluorescent ROS probe that can passively permeate across membranes; (3) 4,5-diaminofluorescein diacetate (DAF-2 DA) and 4-amino-5-methylamino-2′, 7′-difluorofluorescein diacetate (DAF-FM DA) highlight nitric oxide radicals. The primary limitations of these techniques include a great reactivity of ROS and the lack of a standardized approach [24]—therefore measuring biomarkers based on oxidative stress-induced modification (such as for protein, lipid, and DNA damage) becomes a relevant issue.

Among these biomarkers, malondialdehyde (MDA) and thiobarbituric acid reactive substances (TBARS) are commonly used to evaluate lipid peroxidation products [25,26]. As a result, further benchmarks, such as 4-hydroxy-2-nonenal (4-HNE), conjugated dienes (CD), lipid hydroperoxides (LOOH), and 8-isoprostaglandin F2α (8-iso-PGF2α), produced by arachidonic acid peroxidation, were introduced to evaluate lipid peroxidation in biological fluids samples (e.g., urine and plasma) [27]. Moreover, several methods have been developed to evaluate protein oxidative modification, such as determining advanced oxidation protein products (AOPP) [28,29]. Besides lipids and proteins, even DNA double strands undergo chemical modification that can determine genetic damages on the daughter strands [14,30,31]. Finally, analysis of oxidative stress has been carried out, estimating levels and activities of enzymatic and non-enzymatic antioxidants in biological samples, such as plasma, serum, and tissue samples. More specifically, superoxide dismutase (SOD), catalase, glutathione peroxidase (GPx) and glutathione S-Transferase (GSTs), H_2_O_2_ activity (HBA) are taken into account in the determination of enzymatic antioxidant status [3,32].

Recently, there is attention to the validation of new biomarkers of oxidative stress, as they have a potential application in clinical practice. According to the World Health Organization, a biomarker is “any substance structure or process that can be measured in the body or its products and influence or predict the incidence of outcome or disease” [33].

Among the biomarkers of oxidative stress used in the clinical practice, we can find:

### 2.1. Advanced Glycation End Products (AGEs)

AGEs are a class of heterogeneous molecules that are the result of a series of non-enzymatic reactions (Maillard reaction) between reducing sugars and proteins amino groups [34]. AGEs are normally present in the organism, but in oxidative stress conditions, their concentration is higher [35]. When this happens, they interact with a specific receptor on endothelial cells (EC), called RAGE, and promote changes in EC and pericytes typical of diabetes complications, for example. AGEs exercise growth inhibitory and toxic actions on pericytes; RAGE also promotes leukocyte adhesion to EC and monocyte migrations. All these effects cause inflammation, which can aggravate diabetes vascular complications [36]. Furthermore, elevated levels of AGEs have been associated with numerous conditions, including aging [37], neurodegenerative diseases [38], obesity, diabetes mellitus, human immunodeficiency virus (HIV), anemia, and hepatocellular carcinoma [39].

### 2.2. Oxidized Low-Density Lipoprotein (oxLDL)

The measurement of oxLDL as a biomarker of oxidative stress has its origin in the oxidative modification hypothesis of atherosclerosis [40]. Many studies demonstrated that ox-LDL induces inflammatory reactions by activating many types of cells in the vascular wall, including macrophages, smooth muscle cells, and endothelial cells, which suggests that they are a strong factor in the progression and formation of the atherosclerotic lesion. The systemic oxLDL serum levels could represent a biomarker of oxidative stress for early diagnosis and faster initiation of treatment in CVDs.

### 2.3. Protein Oxidation Is Advanced Oxidation Protein Products (AOPP)

AOPP levels are a measure of highly oxidized proteins, especially albumin, and its plasma levels correlate with plasma concentrations of AGE [41]. Furthermore, plasma levels of AOPP were elevated in a different pathological setting like coronary artery disease (CAD) [42], and diabetes [43].

### 2.4. Lipid Oxidation Products

Polyunsaturated fatty acids (PUFAs), in particular linoleic and arachidonic acid (AA), are the main targets of lipid peroxidation. The reaction of ROS with these molecules initiates the autocatalytic reaction of lipid peroxidation, during secondary products are formed, such as Isoprostanes (IsoPs), malondialdehyde (MDA), and trans-4-hydroxy-2-nonenal (4-HNE) [44]. IsoPs, and in particular F_2_-IsoPs, are considered one of the best oxidative stress biomarkers [45]. High levels of F2-IsoP are associated with CVD, and particularly correlated with the degree of diseases and predict the outcome [46]. MDA and HNE are extensively utilized biomarkers, used in both in vivo and in vitro studies to predict the onset of many diseases, such as diabetes, hypertension, cancer, heart failure, and atherosclerosis [47]. Frequently, MDA levels were measured as thiobarbituric acid reactive substances (TBARS). The TBARS assay has been applied as an indicator of oxidative stress in association with the measurement of MDA in several cardiovascular disease models. In fact, serum levels of TBARS were elevated in the serum of cigarette smokers [48] and in patients with documented CAD [49]. Thus, levels of TBARS could predict major cardiovascular events and carotid atherosclerotic plaque progression [50].

### 2.5. 8-Hydroxy-2′-Deoxyguanosine (8-OHdG)

8-OHdG derived from a modification due hydroxyl radical attack of deoxyguanosine residues. It has been commonly chosen as a biomarker most representative product of oxidative modifications of DNA, and it is the best non-invasive biomarker of oxidative damage to DNA. High levels of 8-OHdG in blood and urine are significantly associated with both coronary artery disease and other types of atherosclerotic processes, such as stroke, peripheral artery disease, and carotid atherosclerosis [51,52].

### 2.6. Hydrogen Peroxide (H_2_O_2_)

H_2_O_2_ is the main redox metabolite with important physiological roles in cellular processes like membrane signal transduction, gene expression, cell differentiation, insulin metabolism, cell shape determination, and growth factor-induced signaling cascades. However, when produced in excess, cellular H_2_O_2_ plays an important role in the atherothrombotic process, as it can induce the formation of oxidized low-density lipoproteins (oxLDL) and to stimulate platelet activation [32].

### 2.7. NOX2 Activity (sNOX2-dp)

The production of H_2_O_2_ is closely associated with the activity of the enzyme NADPH oxidase, the body’s major producer of ROS. As previously demonstrated, NADPH oxidase, and in particular, its isoform NADPH oxidase 2 (NOX2), after stimulation, produces ROS and releases a small peptide, defined as soluble NOX2-derived peptide (sNOX2-dp) [53]. NOX2-derived peptide (sNOX2-dp) is detected by ELISA method [54,55]. Specifically, the higher sNOX2dp levels are associated with a significantly increased cumulative incidence of cardiovascular events and deaths in patients with atrial fibrillation [56].

The large diversity in biomarkers of oxidative stress in different diseases and pathological conditions makes it essential to choose the best biomarkers of oxidative stress in each specific disease. Particular attention should be paid to biomarkers that evaluate the activity of ROS-generating enzymes, such as NADPH oxidase. Experimental and clinical studies have shown that NOX2 activity, an isoform of NADPH oxidase, is implicated in the main mechanisms of cardiovascular pathology, which is clotting system and platelet activation; therefore, NOX2 inhibition may reduce thrombosis-related vascular disease. It has been demonstrated that NOX2 activity is significantly associated with platelet activation in vivo. In particular, platelet activation was reduced in subjects with different low rates of NOX2 activity, namely, X-linked chronic granulomatous disease (X-CGD, low rate) patients and X-CGD carriers (medium rate) compared to obese patients (high rate of NOX2 activity). These results suggest that upstream inhibition of oxidative stress by targeting precise cellular oxidant pathways, such as NOX2 or other pro-oxidant enzymatic pathways, may represent an alternative option not only to inhibit platelet activation, but also to retard atherosclerotic progression.

## 3. Antioxidants Supplementations: Which Are the Most Effective in Clinical Practice?

The term “antioxidants” defines chemical substances that slow down the damage caused by oxygen to organisms. Antioxidants are one of the mechanisms that the body uses to fight against oxidative stress with the role to balance the negative effects of oxidant agents and protect cells from oxidative damage [2]. We can identify two macro groups of antioxidants: Those who are produced by the body itself (i.e., endogenous antioxidants) and those that derive from dietary sources (i.e., exogenous antioxidants). Endogenous antioxidants can be divided into two classes: Enzymatic and non-enzymatic antioxidants. Some enzymatic antioxidants are catalase (CAT) that degrades hydrogen peroxide (H_2_O_2_) to water and oxygen, glutathione reductase (GRx), glutathione peroxidase (GPx) that catalyzes the reduction of H_2_O_2_ by the reduced form of glutathione (GSH), creating a glutathione bridge with another glutathione molecule (GSSG), and superoxide dismutase (SOD) that catalyzes the dismutation of superoxide anion radical (O_2_^−^) into H_2_O_2_ and oxygen (O_2_) [57].

The non-enzymatic antioxidants include nutrients that are not produced by the body, and thus need, to be included through the diet. Nutrient antioxidants are found in fruits, vegetables, and fish, and are extremely important because each one of them has a role in oxidative stress neutralization [58,59]. According to their role in reducing oxidative stress-mediated cardiovascular risk, these exogenous molecules can represent a useful tool in clinical practice [60]. Specifically, natural extracts, such as polyphenols, exert an antioxidant activity that includes suppression of ROS formation by either inhibition of enzymes involved in their production, like NOX2 [61], scavenging of ROS [62], or up-regulation or protection of antioxidant defenses [63].

The most widely used antioxidants include:

### 3.1. Vitamins E and C

Vitamin E is a strong antioxidant, is dissoluble in fat, and presents eight stereoisomers. Just one, α-tocopherol, is bioactive in humans. The main function of vitamin E is to protect the body against lipid peroxidation. It has been shown that high-dosages (≥400 IU/day or more for at least 1 year) can be dangerous and can increase the risk of death. Moreover, a dose-response analysis showed a statistically significant relationship between vitamin E dosage and all-cause mortality, with an increased risk of dosages greater than 150 IU/day [64]. The effect of Vitamin E supplementation in the prevention of cardiovascular diseases is controversial. The analyses of sixteen randomized controlled trials of vitamin E treatment showed that, compared to controls, vitamin E given alone significantly decreased myocardial infarction (R.R.: 0.82; 95% C.I., 0.70–0.96; *p* = 0.01) [65]. Supplements containing vitamin E significantly reduced cardiovascular mortality risk (RR: 0.88; 95% CI: 0.80, 0.96) [66]. However, the analyses of 15 trials reporting data on 188,209 participants showed that antioxidant vitamin supplementation (vitamin E, β-carotene, and vitamin C) has no effect on the incidence of major cardiovascular events, myocardial infarction, stroke, total death, and cardiac death [67].

Vitamin C, or ascorbic acid, is a water-soluble antioxidant with a fundamental role in quenching various ROS and reactive nitrogen species (RNS). The antioxidant activity of vitamin C supplementation resulted in positive effects when administrated in concentrations that ranged from 500 to 2.000 mg/day. In the case of high consumption, vitamin C and its metabolites, such as dehydroascorbic acid, 2,3-diketogulonic acid, and oxalic acid, are excreted via the kidneys in humans. Vitamin C is generally non-toxic, but at high doses (2–6 g/day) it can cause gastrointestinal disturbances or diarrhea. However, these side effects are generally not serious and can be easily reversed by reducing its intake [68]. Several lines of evidence suggest that Vitamin C may be associated with a favorable impact on the risk of cardiovascular disease. Vitamin C dose greater than 500 mg/day was associated with beneficial effects on endothelial function with stronger effects in those at higher cardiovascular disease risk, such as in atherosclerotic, diabetic, and heart failure patients [69].

The analyses of thirteen trials involving 1956 patients after cardiac surgery showed that vitamin C significantly reduced the incidence of postoperative atrial fibrillation (RR: 0.68, 95% CI: 0.54, 0.87, *p* = 0.002) and the risk of adverse events (RR: 0.45, 95% CI: 0.21, 0.96, *p* = 0.039) [70].

Finally, the effects of Vitamins E and C are strictly correlated. Indeed, in patients with coronary artery disease, supplementation with 2 g of vitamin C with 600 mg of vitamin E orally significantly enhanced endothelium-dependent vasodilatation in the radial circulation [71].

### 3.2. Omega-3 and Omega-6 Fatty Acids

These kinds of fatty acids, characterized by a long aliphatic chain, are essential for human health. They cannot be synthesized, so they must be taken through food. Omega-3 fatty acids are divided into three different types: Eicosapentaenoic acid (EPA), docosahexaenoic acid (DHA), and alpha-linolenic acid (ALA). The EPA and DHA are present in fish and can be used by the body without been changed. ALA, which is present in large quantities in nuts, must be converted to EPA and DHA [72]. Omega-3 fatty acids are involved as an anti-inflammatory countering the process of chronic diseases. Although the ideal amount to take is not firmly established, evidence from prospective secondary prevention studies suggests that intakes of EPA + DHA ranging from 0.5 to 1.8 g per day (either as fatty fish or supplements) significantly reduce the number of deaths from heart disease. Intervention trials with omega-3 fatty acid supplements have reported no serious adverse reactions at the doses administered. The more common adverse effects of fish oil preparations, particularly in higher dosages, include nausea, fishy belching, and loose stools. Moreover, the administration at high doses has been shown to prolong bleeding time [73].

There are much clinical evidence supporting the beneficial effects of EPA and DHA supplementation on cardiovascular health. Treatment of patients with acute myocardial infarction with four 1g capsules per day containing ethyl esters of EPA (465 mg) and DHA (375 mg) was associated with a reduction of adverse left ventricular remodeling, non-infarct myocardial fibrosis, and serum biomarkers of systemic inflammation [74]. In patients with acute coronary syndrome assigned to receive 1800 mg/day of EPA after PCI, death from a cardiovascular cause were significantly reduced [75]. In adults at high cardiovascular risk, omega-3 fatty acids (1800 mg/day for 12 weeks) administration improved arterial stiffness and endothelial function [76]. The supplementation with omega-3 ethyl-ester (1.86 g of EPA and 1.5 g of DHA daily) to subjects with stable coronary artery disease attenuates the fibrous plaque progression compared to placebo [77]. The administration of 2 g twice daily of icosapent ethyl, which is a highly purified and stable EPA ethyl ester, to patients with established cardiovascular disease or with diabetes and other risk factors, significantly reduced the risk of ischemic events, including cardiovascular death compared to placebo [78]. Moreover, in statin-treated patients at increased cardiovascular risk, icosapent ethyl 4 g/day significantly reduced triglycerides, total cholesterol, oxidized LDL, hsCRP, and other atherogenic and inflammatory parameters [79]. The analyses of fourteen randomized controlled trial (71,899 subjects) showed an 8.0% lower risk for cardiac death in long-chain omega-3 polyunsaturated fatty acids arms versus controls [80].

For ALA, a total intake of 1.5 to 3 g per day seems beneficial, although definitive data from prospective, randomized clinical trials are still needed [81].

### 3.3. Polyphenols

Polyphenols are natural compounds synthesized exclusively by plants with chemical features related to phenolic substances. Epidemiological studies suggest that diets rich in polyphenols may be associated with reduced incidence of cardiovascular disorders, due to their antithrombotic, anti-inflammatory, and anti-aggregative properties [82]. Polyphenols can be simply classified into flavonoids and non-flavonoids.

### 3.4. Non-Flavonoids

Non-flavonoids include phenolic acids, stilbenes, and lignans. Among non-flavonoids, resveratrol is a stilbenoid that exhibits a plethora of therapeutic benefits, including anti-inflammatory and antioxidant properties, anti-platelet, anti-hyperlipidemic, immuno-modulator, cardioprotective, vasorelaxant, and neuroprotective effects [83]. It has been shown that doses of resveratrol lower than 0.5 g per person may be sufficient to decrease blood glucose levels, improve insulin action, and generate cardioprotective effects and other favorable effects [84]. A review of the research on resveratrol in the last 10 years showed that a repeated and moderate administration of resveratrol is better than the administration of a single, higher dose. A safe and efficient dose is 1 g or more per day; however, resveratrol intake is safe at a dose of up to 5 g [85].

### 3.5. Flavonoids

Flavonoids, a family of polyphenolic compounds, are potent antioxidants present in most plants and are classified into seven classes. They can be divided into several subgroups corresponding to different classes of plants, which have multiple effects on the human body [86]. There are thousands of flavonoids that can be found in plants in different amounts and combinations. At this time, the totality of evidence suggests long-term consumption of flavonoid-rich foods may be associated with a lower risk of fatal and non-fatal ischemic heart disease (IHD), cerebrovascular disease, and total CVD [87]. The toxicity of flavonoids is very low. However, as a precaution, doses less than 1 mg per adult per day have been recommended for humans [88]. At higher doses, flavonoids may act as mutagens, pro-oxidants that generate free radicals, and as inhibitors of key enzymes involved in hormone metabolism [89].

### 3.6. Carotenoids

β-carotene is a member of the carotenoids, a family of provitamins that can be converted into vitamin A and are naturally found in abundance in vegetables and fruits. Carotenoids are strong antioxidants as they can scavenge the free oxygen radicals from the body. Moreover, carotenoids with oxygen in the structure like fucoxanthin and astaxanthin have proved to suppress the expression of cytokines IL-6, TNF-α, and IL-1β and act like pro and anti-inflammatory compounds [90]. Several epidemiological reports have shown a correlation between elevated dietary carotenoid intake and the prevention of CVD [91]. A safer profile for non-provitamin A carotenoids (up to 20 mg/day for lutein and 75 mg/day for lycopene) and 2–4 mg/day β-carotene has been suggested [92]. However, for β-carotene, serious adverse effects have been reported in large-scale prospective randomized trials: Four years of supplementation with 20 to 30 mg β-carotene per day was associated with increased risk of lung cancer and cardiovascular disease among smokers and workers exposed to asbestos [93].

### 3.7. Selenium

Selenium is an essential dietary mineral that can be found in very low concentrations in seafood, meat, soil, some vegetables, and liver. Selenium is a cofactor of enzymes, such as glutathione peroxidase (GSH-Px), which is a potent antioxidant enzyme. The recommended dietary allowance for selenium that is estimated to be sufficient to meet the nutritional needs of nearly all healthy adults is 55 μg/day. Selenium toxicity can occur with acute or chronic ingestion of excess selenium. An excess of selenium in the diet (>400 μg/day) will result in selenosis, i.e., poisoning by selenium. Symptoms of selenium toxicity include nausea, vomiting, nail discoloration, brittleness, hair loss, fatigue, irritability [94].

The cardioprotective effect of selenium is still controversial, probably due to the limited trial evidence that is available to date. In observational studies, a 50% increase in selenium concentrations was associated with a 24% reduction in coronary heart disease risk [95]. In a clinical study, patients with congestive heart failure, 200 µg/day of selenium for 12 weeks had beneficial effects on insulin metabolism, and markers of cardio-metabolic risk [96]. However, the meta-analyses of twelve trials that included 19,715 participants randomized to selenium supplementation showed that were no statistically significant effects of selenium on all-cause mortality, CVD mortality, or all CVD events (fatal and non-fatal) [97].

### 3.8. Lipoic Acid

Lipoic acid is an organosulfur component produced from plants, animals, and humans. It has a dual role in the body as it is an antioxidant and a cofactor for enzymes involved in the 2-oxoglutarate dehydrogenase complex. It is synthesized by the human at a low number, but the quantities produced are not enough to fulfill the energy requirement of the cell. Thus, it is mostly obtained from the diet, especially from meat, vegetables, and fruits. The lipoic acid in humans, supplemented at the therapeutic range from 200 to 1800 mg/day, has numerous clinically valuable properties. For example, studies supported the potential use of lipoic acid in diabetes, as the major risk factor for developing several human diseases, including atherosclerosis, hypertension, heart failure, and myocardial infarction [98].

### 3.9. Coenzyme Q10

Coenzyme Q10 a naturally occurring, lipid-soluble, vitamin-like substance involved in the mitochondrial electron transport chain, and it is, thus, essential to produce energy in the body. It is essentially present in the heart and in the liver, and it can be assimilated through meat, some fruit and vegetable, and soybean [99]. The risk assessment for CoQ10, based on various clinical trial data, indicates that the safety level is 1200 mg/day/person suggesting that CoQ10 is highly safe for use as a dietary supplement [100]. Recent data indicate that Coenzyme Q10 has an impact on the expression of many genes involved in metabolism, cellular transport, transcription control, and cell signaling, making CoQ10 a potent gene regulator. Therefore, coenzyme Q10 supplementation is useful in diseases associated with CoQ10 deficiency, which includes diabetes mellitus, mitochondrial diseases, and cardiovascular disease [101]. Patients with moderate to severe heart failure randomized to CoQ10 (300 mg daily) in addition to standard therapy, after two years showed reduced major adverse cardiovascular events, all-cause mortality, cardiovascular mortality, hospitalization, and improvement of symptoms [102]. The daily dosage of CoQ_10_ supplement ranged from 60 to 300 mg also resulted in a net increase in ejection fraction of 3.67% (95% CI: 1.60%, 5.74%) in patients with congestive heart failure [103]. The analyses of eight trials (267 participants) showed that taking CoQ10 by patients with CAD significantly decreased total-cholesterol and increased HDL-cholesterol levels [104].

The choice of the type of antioxidant supplementation that best affects cardiovascular disease is still a challenge. The results of several antioxidant supplementations in different cardiovascular diseases are disparately ranging from possibly beneficial to many futile to some harmful effects. The different results may be due to several reasons, including the different concentrations used—also taking into account that high concentrations have negative effects. Moreover, for some supplements, there are no clinical data or data relating to small trials, so it is of importance investigating patient-relevant outcomes.

## 4. Biomarkers of Oxidative Stress in Patients with Cardiovascular Risk Factors

Cardiovascular Disease (CVD) is worldwide known to be a major cause of death and comorbidity. Atherosclerosis is the key pathophysiological mechanism underlying the development of CVD [105]. In particular, atherosclerosis, a chronic inflammation that affects arteries, may remain clinically undetected for many years before an acute event, such as Ischemic Heart Disease (IHD) or a stroke and Peripheral Vascular Disease (PVD) [106].

CVDs are caused by multiple factors that can be divided into un-modifiable and modifiable risk factors. Age, gender, family history, and ethnicity are all un-modifiable because the individual can do nothing to avoid these risk factors.

Though the characteristics of un-modifiable risk are greatly suitable for risk stratification, the modifiable factors have the advantage of being a possible target for pharmaceutical intervention to lower cardiovascular risks. Among the main modifiable traditional cardiovascular risk factors, there are hypertension, diabetes mellitus, obesity, hypercholesterolemia, and smoking (Figure 1). Furthermore, these cardiovascular risk factors are associated with increased production of oxidative stress (Figure 1). Clinical human studies have supported the association between oxidative stress and cardiovascular events, and different types of molecular biomarkers provide a powerful approach to the understanding of cardiovascular risk factors with consequent applications in epidemiology and clinical studies and in the prevention, diagnosis, and management of cardiovascular disease (Table 2).

### 4.1. Hypertension

Many epidemiological studies, like the Framingham study, report that hypertension is directly associated with cardiovascular risk and systolic and diastolic blood pressure is associated with cardiovascular outcomes. Hypertension is one of the most common cardiovascular risk factors, and oxidative stress is important in the molecular mechanisms associated with it [162]. Oxidative stress was observed among hypertensive patients as depicted by the high plasma levels of oxLDL [107,114,119], and reduced enzymatic antioxidant activity, which is determined by glutathione peroxides (GPx) [110,114], total antioxidant capacity (TAOC) [113], ferric reducing ability of plasma (FRAP) [113], total antioxidant capacity (TAC) [115,119]. There are many studies focusing on the comparison of oxidative stress biomarkers in hypertensive patients. Indeed, the pathology of hypertension is characterized by decreased levels of GSH, GPx, and SOD-1, and higher levels of TBARS and MDA [107,109,110,112]. The alteration of these parameters might indicate a condition of more severe oxidative stress compared to control subjects. Moreover, increased MDA levels and reduced SOD activities might be considered as prognostic markers of developing organ damage in patients with hypertension [108]. Moreover, high concentrations of F2-isoprostanes are associated with hypertension and are used to evaluate oxidative stress in this disease [108,113,114]. More studies are warranted to explore possible associations between oxidative damage, antioxidant status and hypertension (Table 2).

### 4.2. Diabetes

Some studies show that diabetes, a metabolic disorder characterized by a high blood sugar level over a prolonged period, is correlated with oxidative stress and with a 2- to 3-fold increase in the likelihood of developing CVD [163]. Focusing on concentrations of oxidative biomarkers, such as F-2 isoprostanes and ox-LDL in plasma samples, it has been demonstrated that they were positively associated with T2DM [122]. In the same way, in urine samples of patients with cardiovascular risk, oxidative stress biomarkers, such as F2-IsoP [118], increased with the increase of CVD risk. Case-control studies suggest that, in patients with T2MD, serum concentrations of oxidative stress biomarkers, such as MDA, GSSG were increased in T2MD patients, conversely, GSH levels were lower compared to controls [124]. Moreover, in urine samples of diabetic patients with the acute coronary syndrome (ACS), higher 8-iso-prostaglandin F2α (8-iso-PGF2α) levels were correlated with increased necrotic plaque components [126]. Moreover, Anderson et al. measured MDA concentration and TAC levels in plasma samples of women with gestational diabetes (GDM), and both parameters resulted higher and lower, respectively [121]. Finally, it was demonstrated that NOX2 can contribute to the formation of 8-iso-PGF2α in both platelets and urine [128]. These biomarkers were found higher in diabetic versus non-diabetic patients [129]. This highlights the hypothesis that NOX2 is crucial for the ROS overproduction observed in T2DM [164], and suggests a role for NOX2 in platelet isoprostanes overproduction in T2DM (Table 2).

### 4.3. Hypercholesterolemia

Hypercholesterolemia is implicated in complications and increases in cardiovascular risk. Indeed, there is a direct and positive correlation between total cholesterol and low-density lipoprotein cholesterol (LDL-C) plasmatic levels and CVD risk. Many epidemiological studies highlight total cholesterol could be a useful marker for predicting CVD [162]. The importance of the relationship between LDL-C level and CVD risk is confirmed by the efficacy of LDL-C lowering drug therapies that are able to reduce event and mortality CVD related [165]. Additionally, hypercholesterolemia is a condition closely correlated with oxidative stress condition. A cross-sectional, observational study involving 132 patients with familial hypercholesterolemia (FH) reported enhanced oxidative stress in FH subjects compared to normolipidemic subjects. Indeed, according to the reference range (>1.24 g/L) for MDA based on the International Federation of Clinical Chemists (IFCC), MDA concentration is greatly elevated in FH, and this suggests enhanced oxidative stress status [130]. High levels of plasma and serum MDA were also reported by other researchers [132,133]. MDA is a reflection of lipid peroxidation, and several studies have shown these mechanisms in hypercholesterolemia by measurement of oxLDL [130,131,134,138,140]. HC showed an increased platelet activation with consequent thrombus formation [166]. In particular, Barale et al. demonstrated that, in FH, platelet reactivity is correlated with biomarkers of redox function, including SOD and the in vivo marker of oxidative stress urinary 8-iso-prostaglandin F2α [167]. In addition, many authors observed, in FH in parallel to the high levels of LDLc, reduced GSH, SOD, and CAT levels, and increased ROS production [130,135,136] (Table 2).

### 4.4. Obesity

Obesity is characterized by an increase in body weight and represents a social problem worldwide. Obesity is associated with various comorbidities, including CVD, and it is also characterized by chronic low-grade inflammation associated with increased oxidative stress. Oxidative stress damage leads to the development of obesity-related complications [168]. The combination of excess production of ROS and inefficient antioxidant capacity generates oxidative stress, which represents one of the main mechanisms underlying the development of obesity [141]. Many studies demonstrated enhancement of the antioxidant barrier (SOD, CAT, GPx, and TAC [141,148,149,151,153]) with a simultaneous decrease of glutathione. The concentration of the products of oxidative damage to proteins (AGE) [151], lipids (MDA) [141,148,151], and DNA (8-OHdG) [142,147,151], as well as total oxidative status, were significantly higher in both saliva and plasma of overweight and obese subjects. The measure of some markers like catalase and TAC could be used to assess the central antioxidant status of overweight and obese [151]. In hypercholesterolemic subjects, the higher level of oxidative stress could be due to a higher concentration of SOD [143,148,151]. Finally, Loffredo et al. analyzed the interplay among oxidative stress, NOX2, the catalytic core of NADPH oxidase in children with obesity and/or hypercholesterolemia. The results showed that oxLDL, urinary excretion of 8-iso-PGF2α and NOX2 activity, as assessed by serum levels of sNOX2-dp, were higher in obese children than in control groups, suggesting that NOX2-generating oxidative stress may have a pathogenic role in the functional changes in obesity and hypercholesterolemia [138] (Table2).

### 4.5. Smoking

Smoking, according to the statistics, has devastating consequences on individuals and society. It is estimated that by 2025, there will be 1.6 billion smokers in the world, and 10 million people a year will die from smoking [169,170]. Smoking increases the overall risk of CVD through different mechanisms, such as endothelial dysfunction, inflammation leading to atherosclerosis, dyslipidemia, and oxygen demand-supply mismatch in the myocardium [171,172]. In addition, cigarette smoke can induce the overproduction of ROS by many of the cellular enzyme systems. Thus, the induction of oxidative stress, exacerbated by cigarette smoke, decreases protection due to antioxidants systems [173].

There is large evidence linking cigarette smoke with the induction of oxidative stress and the onset and progression of major smoking-related diseases, such as cardiovascular disease, cancer, and chronic obstructive pulmonary disease (COPD) [174]. The urinary metabolite 8-iso-PGF2α is an accepted biomarker of oxidative damage. In fact, many studies showed that in adult smokers, levels of 8-epiPGF2α were significantly higher than non-smokers [155,158,159]. Moreover, compared with healthy subjects, smokers showed enhanced levels of oxidative stress, measured as ROS production, NOX2 activation, and 8-iso-PGF2α formation [61,160,161].

Furthermore, some studies analyzed the antioxidant systems smoking-related by the analysis of oxidative stress biomarkers, including lipid hydroperoxide (LOOH), total antioxidant status (TAS), total oxidant status (TOS), oxidative stress index (OSI), paraoxonase (PON). In this regard, young smokers diagnosed with acute myocardial infarction, had significantly higher OSI and TOS levels and lower TAS and LOOH levels. Moreover, CAD severity correlated positively with OSI and TOS levels, suggesting that high levels of OSI and TOS could be considered as indicators of disease severity and heavy smoking-related vascular damage in early-onset CAD [119,157]. Finally, Frati et al. showed that smokers had an increase in oxidative stress markers and a worsening of antioxidant systems compared to non-smoker [161] (Table2).

## 5. Antioxidant Supplementation in Patients with Cardiovascular Risk Factors

As described in the previous paragraph, oxidative stress characterizes several cardiovascular patients, such as those with diabetes, obesity, hypercholesterolemia, and hypertension, but also smokers. In the following chapter, the role of the supplementation with antioxidants, to counteract oxidative stress and correlated damages, will be discussed. In particular, we reviewed in human randomized clinical trials published over the past 10 years, the effect of micronutrient supplementation (Table 3).

### 5.1. Hypertension

The pathophysiology of hypertension involves a complex interaction of multiple vascular effectors, including the activation of the sympathetic nervous system, of the renin-angiotensin-aldosterone system and the inflammatory mediators. Oxidative stress and endothelial dysfunction are consistently observed in hypertensive subjects and have a causal role in the molecular processes leading to hypertension.

Tousoulis et al. evaluated the effect of high-dose vitamin C and vitamin E on endothelial function in hypertensive patients. They found that vitamin pre-treatment failed to prevent methionine-induced homocysteinemia reduction of endothelium-dependent dilation despite the reduction of peroxidation induced by vitamins [175]. In another study, in hypertensive patients, the antioxidant vitamins C and E supplementation resulted in a reduction in blood pressure, oxidative stress biomarkers, and increased fluidity by PUFA proportion in the membrane. Reduction of oxidative stress and changes in membrane fluidity positively modulates (Na, K)-ATPase activity accounting for the blood pressure reduction [176].

In pre-hypertensive men and women, the effect of quercetin and epicatechin on the concentrations of methylglyoxal (MGO) and advanced glycation end products (AGEs) was evaluated. The results showed that quercetin, but not epicatechin decreased the plasma concentration of MGO, which is a reactive di-carbonyl intermediate and a precursor of AGEs [177]. On the same line, Saarenhovi et al. found a significant acute improvement in maximum FMD% after epicatechin supplementation, but not statistically significant compared to placebo [178].

Few studies evaluated the effect of Coenzyme Q-10 supplementation. In mildly hypertensive patients, coenzyme Q10 supplementation was effective in decreasing some pro-inflammatory factors, such as IL6 and hs-CRP. Moreover, coenzyme Q10 increased adiponectin levels, an adipokine with anti-inflammatory and anti-atherogenic effects [216] that may be involved in the progression of hypertension [179]. Adjunctive coenzyme Q10 therapy was not associated with statistically significant reductions in systolic or diastolic blood pressure or heart rate [180] (Table 3).

### 5.2. Diabetes

As oxidative stress plays a key role in the development and the progression of diabetes and its related complications, several antioxidant supplementations were tested.

The supplementation with antioxidant vitamins in diabetic patients exerts beneficial effects that could improve the clinical condition and attenuate or prevent diabetic pathogenesis and complications that, secondly to poor glycemic control, could attribute to the imbalance between the decline in the endogenous antioxidants and increasing production of the ROS. Indeed, Vitamin C or Vitamin E supplementation improves fasting blood sugar (FBS), lipid profile, insulin, homeostasis model assessment of insulin resistance (HOMA-IR) [182,184], and then increases the antioxidant profile and reduces oxidative biomarkers [182,184,185,217]. However, other studies do not support the beneficial effect of vitamins supplementation. For example, no differences were seen in the endothelial function measurement and total plasma antioxidant capacity (TAOC) before and after combined vitamin C and E therapy [181] or in plasma oxyphytosterol concentrations and other oxidative biomarkers [183].

The antioxidant effects of resveratrol supplementation in attenuating the increased oxidative stress in diabetes mellitus patients have been investigated in several studies [186,187,188,189,190].

In patients supplemented with resveratrol, glycemic indices, such as fasting blood sugar, HbA1c, insulin levels, and insulin resistance were all significantly decreased in the resveratrol compared with the placebo group [186,190], and improves arterial stiffness as indicated by decreased cardio-ankle vascular index, which is a clinical surrogate marker of atherosclerosis [189]. The mechanisms behind these metabolic effects might be due to a resveratrol-induced decrease in oxidative stress. Indeed, after the supplementation with resveratrol, diabetic patients displayed reduced oxidative stress biomarkers as indicated by reduced levels of NO, MDA, and superoxide anion. Moreover, a decreased urinary excretion rate of ortho-tyrosine was observed with resveratrol treatment, indicating a lowered degree of hydroxyl free radical production in these patients [190]. Parallel to the reduction in oxidative biomarkers improved antioxidant status was observed with a significant increase of SOD, GSH-Px, and CAT levels. The antioxidant properties of resveratrol could result from its direct effects by acting as a free radical scavenger, as well as from its ability to indirectly activate antioxidant enzymes and other mechanisms. These indirect effects could be conferred via increased SIRT-1 expression that was associated with significant H3K56ac content reduction and increased serum antioxidant activity in T2DM patients [188]. These findings support the notion that resveratrol decreases oxidative stress through its broad direct and indirect antioxidant effects, and this could be a promising approach for the prevention and treatment of diabetes mellitus.

Only one study, a double-blind randomized crossover controlled clinical trial, evaluated the effect of beta-carotene supplementation. The consumption of beta-carotene fortified synbiotic food resulted in a significant decrease in insulin, HOMA-IR, and a significant increase in plasma nitric oxide and glutathione (GSH) [191]. Likewise, selenium supplements to T2DM patients resulted in a significant decrease in insulin and HOMA-IR contextually to a significant rise in plasma total antioxidant capacity (TAC) concentrations [193].

Several studies evaluated the effect of a supplement containing lipoic acid on glyco-metabolic control and oxidative stress markers. Derosa et al. found that food supplement containing α-lipoic acid reduces fasting plasma glucose, glycated hemoglobin (HbA_1c_), and fasting plasma insulin with an improvement of lipid profile. The antioxidant effect resulted in an increase of SOD, and GSH-Px, and a decrease of MDA [194]. Similarly, Zhao et al. found that α-lipoic acid was safe and effective in treating aged T2DM as blood glucose, lipids, and HOMA-IA of the experiment group decreased significantly. Oxidative stress was affected by supplementation as an increase in plasma SOD and GSH-Px levels, and a decrease in MDA was found [196]. The effectiveness of oral supplementation of alpha-lipoic acid on glycemic status was also confirmed by another study, but with slight efficiency on oxidative stress-related biomarkers [195]. Finally, lipoic acid supplementation did not reduce plasma oxyphytosterol and oxycholesterol concentrations [183] (Table 3).

### 5.3. Hypercholesterolemia

The interaction of the combination of statins with *n*-3 fatty acids on oxidative stress was evaluated in hypercholesterolemic women receiving a mixture of EPA and DHA. Results showed that statins and *n-*3 fatty acids increased oxidative stress as a result of increased plasma malondialdehyde, whereas SOD activity reduced catalase expression [197]. Accordingly, administration of *N*-3 fatty acids to patients treated with statins has no effect on oxidative stress parameter, which is STAT-8-Isoprostane, and on endothelial function. However, combination of statins and *N*-3 fatty acid inhibits platelet aggregation, alters inflammatory status, and positively affects daytime blood pressure [198].

Only one study evaluated the effect of resveratrol in hypercholesterolemic patients. In these patients, with a higher demand for antioxidant activity, due to higher cholesterol levels, resveratrol consumption significantly increased Vitamin E levels without changes in TAC or in total cholesterol levels [199] (Table 3).

### 5.4. Obesity

The effect of antioxidant supplementation on biomarkers of oxidative stress, inflammation, and liver function was evaluated in overweight or obese children and adolescents randomized to intervention with daily antioxidants, namely, vitamin E, vitamin C, and selenium or placebo. Results showed that antioxidant supplementation improved antioxidant-oxidant balance by increasing antioxidant status and reducing oxidative stress biomarkers, namely, F(2)-isoprostanes and F(2)-isoprostane metabolites, but did not affect the inflammatory markers measured [200]. Vitamin C intravenous infusions in overweight or obese grade I subjects reduced protein carbonylation, one of the most harmful irreversible oxidative protein modifications, and a major hallmark of oxidative stress-related disorders [201]. Conjugated linoleic acid supplementation plus vitamin E improved insulin resistance, lipid disturbances, oxidative stress as total antioxidant capacity increased, and MDA significantly decreased in obese patients with NAFLD [202].

Several studies evaluated the effect of resveratrol. Combined polyphenols epigallocatechin-gallate and resveratrol supplemented to obese subjects significantly decreased expression of pathways related to energy metabolism, oxidative stress, inflammation [203]. Accordingly, De Groote et al. demonstrated that resveratrol triphosphate supplementation could contribute to a significant reduction of oxidative stress gene expression. Moreover, daily, chronic resveratrol supplementation maintains healthy circulatory function in obese function as indicated by a 23% increase in FMD compared to placebo [205]. Finally, Wong et al. confirmed the positive effect of resveratrol on vascular function by demonstrating that resveratrol increased FMD in a dose-related manner [206].

Among antioxidant supplementation, several studies focused on lipoic acid effects. Short-term treatment α-lipoic acid supplementation, in obese subjects with impaired glucose tolerance (IGT), improves insulin sensitivity and plasma lipid profile. At the same time, plasma oxidative products, such as MDA and 8-iso-prostaglandin, and inflammatory markers, such as tumor necrosis factor-α (TNF-α) and interleukin-6 (IL-6), remarkably decreased while adiponectin increased [209]. A beneficial effect was also achieved by a combined supplementation of α-lipoic acid, carnosine, and thiamine that was able to reduce glucose and HbA_1c_ levels and a significant reduction in serum hydroperoxide levels [210]. However, McNeilly et al. found that, in obese subjects with IGT, although total oxidant status was lower, α-lipoic acid ingestion may increase the atherogenicity of LDL. When α-lipoic acid is combined with exercise, this atherogenic effect is abolished [207]. Finally, α-lipoic acid administered orally did not protect against lipid-induced insulin resistance in overweight and obese humans. Indeed, after infusion of intralipid plus heparin to raise plasma free fatty acids, insulin sensitivity was impaired even in the case of α-lipoic acid pre-treatment [208] (Table 3).

### 5.5. Smoke

It has been demonstrated that cigarette smoke-induced oxidative stress is responsible for endothelium activation through the expression of adhesion molecules and the activation of macrophages and platelets contributing to endothelial dysfunction. The direct effect of smoke compounds is the ROS overproduction that induces endothelial cell loss through apoptosis or necrosis processes.

The effect of Vitamin E was evaluated in healthy smokers who quit smoking for seven days. The short-term γ-T-rich supplementation, in combination with smoking cessation, improved vascular endothelial function—as indicated by increased brachial artery flow-mediated dilation (FMD) by 1.3%. Moreover, the pro-inflammatory levels of mediators, such as TNF-α and myeloperoxidase, decreased after γ-T-rich supplements, and these were inversely related to FMD. However, the supplementation doesn’t affect plasma oxidized LDL and urinary F2-isoprostanes [211]. In healthy smokers who received nicotine replacement therapy, oral administration of a γ-T-rich mixture of tocopherols increased FMD without affecting plasma nitrate/nitrite. Oxidative stress, as assessed by urinary 8-iso-15(S)-PGF_2α_, decreased in smokers receiving γ-T-rich mixture and was inversely correlated to FMD [212]. Finally, long-term supplementation with vitamin E (36 months) lowered oxidative stress in smokers, as measured by urine 8-iso-prostaglandin F2- α (8-iso-PGF2α) by 21% [213]. In the same patients that smoke, no evidence for an effect was observed for combined vitamin E and selenium or selenium alone intervention [213].

As smoking-induced oxidative stress is thought to contribute to lower levels of omega-3 fatty acids in plasma, Sadeghi-Ardekani et al. evaluated the effects of omega-3 fatty acid supplementation on oxidative stress index in heavy-smoker males. They found that high-dose of omega-3 fatty acid supplements (180 mg of eicosapentaenoic acid and 120 mg of docosahexaenoic acid) for three months decrease total oxidant status and oxidative stress index [214].

One randomized, double-blind, crossover trial was performed to test the hypothesis that resveratrol induces a decrease in the levels of the inflammatory and oxidative mediators that characterize the low-grade systemic inflammatory state and the oxidant-antioxidant imbalance in smokers. The results confirm that oral supplementation of resveratrol for 30 days significantly reduced C-reactive protein (CRP) and triglyceride concentrations, and increased Total Antioxidant Status (TAS) values [215] (Table 3).

## 6. Conclusions

To date, the role of oxidative stress in the onset and progression of atherosclerosis and its impact on the development of cardiovascular events has been widely described. Several studies demonstrated an increase of biomarkers of oxidative stress in the setting of cardiovascular disease and outcome. Even if the suppression of oxidative stress using antioxidants is beneficial, as reported by many clinical studies, the effectiveness of antioxidant therapies is controversial for several reasons. First of all, in different stages of diseases, oxidative stress may have different roles according to the oxidative stress levels. Thus, it is crucial to verify oxidative stress biomarkers levels to give the antioxidant treatments at the appropriate time. For example, researchers or clinicians should focus on the antioxidant power of a patient rather than looking for a particular antioxidant. Among the methods that evaluate this power, HBA is a promising new method that is able to evaluate the ability of each individual to neutralize H_2_O_2_, a cell-permeable ROS generated by cellular metabolism involved in intracellular signaling, which exerts a strong impact on cardiovascular pathophysiology. Indeed, in patients with atrial fibrillation, a reduced ability to scavenge H_2_O_2_, as indicated by reduced serum HBA, predicted cardiovascular events [32].

Moreover, oxidative stress should be assessed with methods that evaluate more the activity of the enzymes involved in the production of ROS than molecules produced by oxidative stress. Among these methods, the evaluation of NOX2 activity seems to correlate well with the severity of cardiovascular diseases and also with cardiovascular events.

Second, not all antioxidants are effective in modifying the outcome of cardiovascular risk factors, and the dosing strategies in clinical trials are different, even in the same pathology. Finally, not all biomarkers of oxidative stress are useful for monitoring the clinical outcome of cardiovascular risk factors. Taken together, this information highlighted that antioxidant therapy must be considered to all intents and purposes a pharmacological therapy, and therefore, it is extremely important to monitor the dosage and time of administration, as suggested in Figure 2.

## Figures and Tables

**Figure 1 antioxidants-10-00146-f001:**
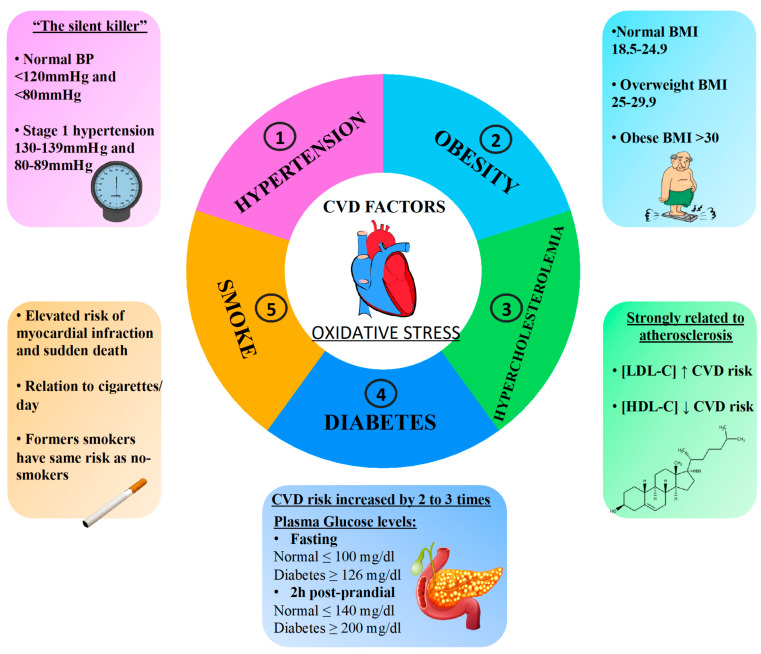
Several cardiovascular risk factors, such as hypertension, obesity, hypercholesterolemia, diabetes, and smoking, are associated with enhanced oxidative stress, which favors the progression of cardiovascular disease.

**Figure 2 antioxidants-10-00146-f002:**
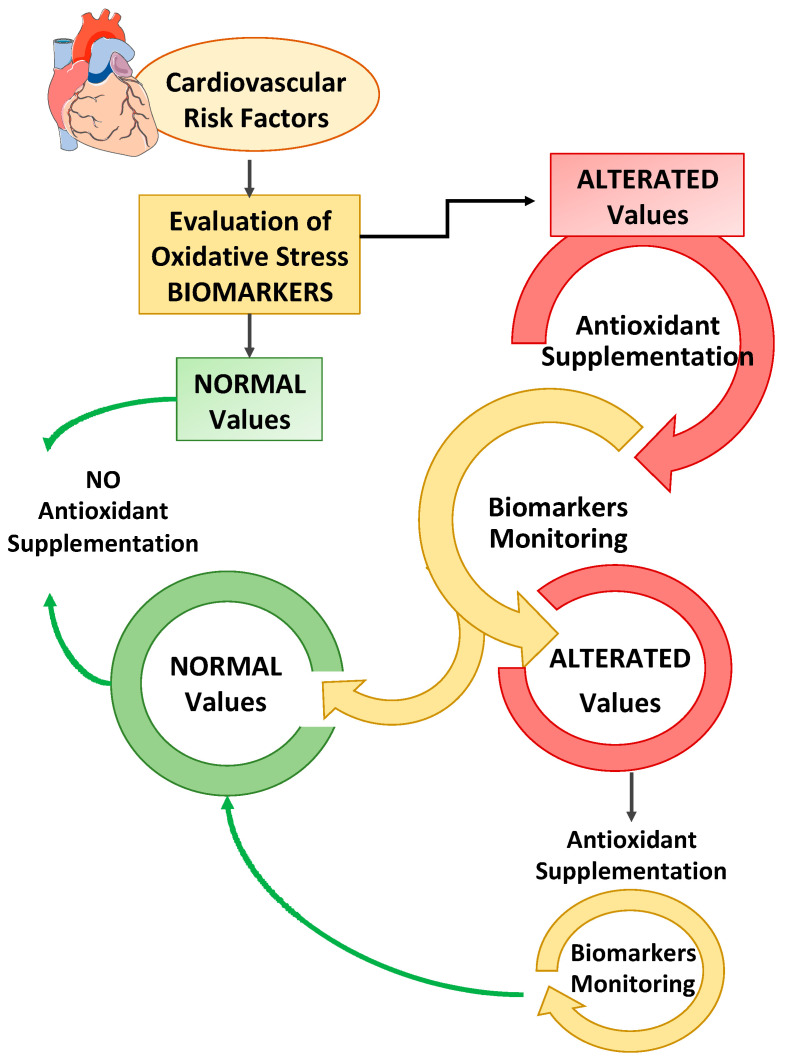
The measurement and monitoring of biomarkers of oxidative stress and antioxidant status associated with cardiovascular risk factors (hypertension, obesity, hypercholesterolemia, diabetes, and smoking) can help (1) to assess the patient’s health status, and (2) to consider an appropriate supplementation of antioxidants if altered oxidative stress/or antioxidant status is recorded. Moreover, antioxidant therapy must be monitored to adjust the dosage and the time of administration based on the biomarker values verified.

**Table 1 antioxidants-10-00146-t001:** Direct and indirect methods to evaluate biomarkers of oxidative stress.

Type of Biomarkers	Direct/Indirect Measurement	Method of Detection	Type of Sample	References
DCFH-DA	Direct	Flow-cytometer	Platelets and leukocytes	[3]
DHR123	Direct	Flow-cytometer	Leukocytes	[3]
DAF-2-DA				
DAF-FM				
D-Rooms	Direct	Flow-cytometer	Serum	[4,5]
C11-BODIPY581/591	Direct	Flow-cytometer	Platelets, leukocytes granulocytes	[6]
4-HNE	Indirect: lipid oxidation	ELISA	Urine and plasma	[7,8]
MDA	Indirect: lipid oxidation	HPLC	Urine and plasma	[9]
TBARS		ELISA		
F2-IsoPs	Indirect: lipid oxidation	Gas-chromatography	Biological fluids	[10,11]
		ELISA		
DNPH	Indirect: protein damage	Colorimetric	Biological fluids	[12,13]
AOPP				
8-OHdG	Indirect: DNA damage	ELISA	Blood and urine	[14,15]
SOD Catalase	Indirect: enzymatic antioxidants	Colorimetric	Biological samples	[16,17]
		Western blots		
GPx		Activity assays		
GSTs				
Endogenous and nutritional elements (glutathione, Vitamins A, C, E)	Indirect: enzymatic antioxidants	Colorimetric, HPLC, Gas-chromatography	Plasma/serum and tissue samples	[18,19]

4HNE = 4-hydroxynonenal; 8-OHdG = 8-Oxo-2′-deoxyguanosine; AOPP = Advanced Oxidation Protein Products; d-ROMS = Reactive Oxygen Metabolites; DCFH-DA = 2′-7′dichlorofluorescin diacetate; DHR123 = Dihydrorhodamine 123; DAF-2 DA = Diaminofluorescein-2 diacetate; DAF-FM = Diaminofluorescein-FM diacetate; MDA = Malondialdehyde; SOD = Superoxide dismutase; TBARS = Thiobarbituric acid reactive substance.

**Table 2 antioxidants-10-00146-t002:** Biomarkers of oxidative stress in patients with cardiovascular risk factors.

Subjects and Healthy Status	Type of Sampling	Type of Biomarkers	References
Hypertension			
N 86	Plasma, Erythrocytes	TBARS	[107]
Children PH		oxLDL	
		GPX and GSH activity	
N 100	Serum	SOD	[108]
		CAT	
		GSH-Px	
		MDA	
		8-iso-PGF2α	
Hypertension			
N 100	Fresh whole blood	TBARS	[109]
Pregnant women with hypertension			
N 49	Serum	GSH	[110]
		GPx	
		SOD	
	Plasma	CAT	
	Erythrocytes	TBARS	
Hypertension			
N 402	Plasma	SOD	[111]
		MDA	
Hypertension		4-HNE	
N 32	Tracheal aspirate	TBARS	[112]
Pregnant with hypertension			
N 150	Serum	8-epi-PGF2α	[113]
		TOAC	
Pregnant with hypertension		FRAP	
N 91	Plasma	GPx activity	[114]
		TAC	
		oxLDL	
		8-epi-PGF2α	
Hypertension			
N 30	Serum	MDA	[115]
Hypertension		TAC	
N 25	Plasma	MDA	[116]
Hypertension		TAC	
N 12	Plasma	MDA	[117]
		GSH	
		Vitamin A and Vitamin E	
Hypertension			
N 897	Urine	8-epi-PGF2α	[118]
Hypertension			
N 54	Serum	TOS/TOC	[119]
		TAS/TAC	
Hypertension		oxLDL	
Diabetes			
N 3766	Plasma	8-oxo-2′-dG	[120]
T2DM			
N 60	Plasma	TAC	[121]
		GSH	
GDM		MDA	
N 2339	Plasma	F2-isoprostanes	[122]
		Carotenoid	
T2D	Serum	Tocopherol	
N 1381	Urine	8-oxoGuo	[123]
T2D			
N 275	Red cell hemolysate	TBARS	[124]
		GSH	
		MDA	
T2D		GSSG	
N 95	Plasma	MDA oxLDL	[125]
T2D			
N 79	Urine	8-iso-PGF_2α_	[126]
T2D			
N 35	Serum	oxLDL	[127]
IGF			
N 121	Urine	8-iso-PGF_2α_	[128]
T2DM	Serum	sNOX2 dp	
N 19	Serum	TOS/TOC	[119]
		TAS/TAC	
Diabetes		oxLDL	
N 50	Urine	8-iso-PGF_2α_	[129]
		sNOX2 dp	
T2DM	Platelets	ROS	
Hypercholesterolemia			
N 131	Serum	oxLDL	[130]
		MDA	
FH		8-iso-PGF_2α_	
N 43	Serum	TBARS	[131]
		oxLDL	
LDL > 160 mg/dL		PON1	
N 24	Plasma	ORAC	[132]
		FRAP	
Hypercholesterolemia		MDA	
N 27	Plasma	ORAC	[133]
		FRAP	
Hypercholesterolemia		MDA	
N 48	Serum	TBARS	[134]
		H_2_O_2_	
Hypercholesterolemia		oxLDL	
N 61	Plasma	SOD	[135]
FH	Urine	8-iso-PGF_2α_	
N 39	Serum	ROS	[136]
		GSH	
		SOD	
Hypercholesterolemia	Plasma	CAT	
N 125	Serum	sNOX2-dp	[137]
Hypercholesterolemia		oxLDL	
N 40	Serum	8-iso-PGF2α	[138]
		sNOX2-dp	
Hypercholesterolemia	Urine	oxLDL	
N 30	Serum	sNOX2-dp	[139]
Hypercholesterolemia	Urine	8-iso-PGF2α	
N 153	Plasma	8-iso-PGF_2α_	[140]
Hypercholesterolemia		oxLDL	
Obesity			
N 10	Plasma	TBARS	[141]
		MDA	
		TAC	
		CAT	
Obese children	Erythrocytes	8-iso-PGF_2α_	
	Urine		
N 88	Urine	8-iso-PGF_2α_	[142]
Obese children		8-OHdG	
N 30	Plasma	GSH	[143]
		GPx	
		SOD	
Obese Adult	Erythrocyte lysate	TAS	
N 20	Plasma	FRAP	[144]
Obese	Urine	Polyphenol content	
N 160	Plasma	4-HNE	[145]
Obese			
N 113	Urine	15-keto-dihydro-PGF2α	[146]
Overweight		8-iso-PGF_2α_	
N 65	Serum	8-OHdG	[147]
Obese		TAS	
N 20	Plasma	MDA	[148]
		SOD	
		GPx	
Overweight and obese adolescents		TAC	
N 75	Serum	TAC	[149]
Obese		ROS	
N not applicable	Urine	TBARS	[150]
Obese		8-iso-PGF2α	
N 40	Salivary	SOD and CAT	[151]
		TAC and TOS	
		GSH	
		AGE	
		MDA	
		8-OHdG and 4-HNE	
Overweight and obese adolescents	Plasma		
N 62	Serum	TOC and TAC	[152]
		OSI	
Obese		oxLDL	
N 27	Serum	TAC	[153]
		TOS	
Obese		OSI	
N 20	Serum	8-iso-PGF2α	[138]
		sNOX2-dp	
Hypercholesterolemia	Urine	oxLDL	
N 35	Urine	8-iso-PGF2α	[154]
Obese			
Smoke			
N 15	Urine	8-*iso*-PGF_2α_	[155]
Smokers			
N 23	Plasma	oxLDL	[156]
Habitual e-cigarette users			
N 33	Serum	LOOH	[157]
		TAS	
		TOS	
		OSI	
		PON	
Smokers			
N 3585	Urine	8-*iso*-PGF2α	[158]
Smokers			
N 20	Serum	sNOX2-dp	[159]
		8-*iso*-PGF2α	
		Vitamin E	
Smokers			
N 20	Serum	sNOX2-dp	[61]
		8-*iso*-PGF2α	
Smokers	Platelets	ROS	
N 20	Serum	sNOX2-dp	[160]
		8-*iso*-PGF2α	
Smokers		Vitamin E	
N 20	Serum	sNOX2-dp	[161]
		8-*iso*-PGF2α	
		H_2_O_2_	
		HBA	
		Vitamin E	
Smokers			
N 25	Serum	TOS/TOC	[119]
		TAS/TAC	
Smokers		oxLDL	

4-HNE = 4-hydroxynonenal; 8-iso-PGF_2α_ = 8-iso-prostaglandin F_2α_; 8-oxoGuo = 8-oxo-7, 8-dihydroguanosine; 8-oxo-2′-dG = 8-Oxo-2′-deoxyguanosine; CAT = catalase; FH = Familial hypercholesterolemia; FRAP = Ferric reducing ability of plasma; GDM = Gestational diabetes mellitus; GPx = glutathione peroxides; GSH = reduced glutathione; HBA = H_2_O_2_ breakdown activity; IFG = Impaired fasting glucose; LOOH = lipid hydroperoxide; MDA = malondialdehyde; NO = nitric oxide; ORAC = Oxygen radical absorbance capacity; OSI = oxidative stress index; ox-LDL = oxidized LDL cholesterol; PAH = pulmonary artery hypertension; PH = primary hypertension; PON = paraoxonase; SOD = superoxide dismutase; T2D= type 2 diabetes (T2D); T2DM = Type 2 diabetes mellitus; TAC = total antioxidant capacity; TAOC = Total antioxidant capacity; TAS = total antioxidant status; TBARS = Thiobarbituric acid reactive substances; TOS = total oxidant status.

**Table 3 antioxidants-10-00146-t003:** Main characteristics and main results of supplementation studies with antioxidants in cardiovascular risk factors.

Groups of Supplementation	Dose		Subjects and Healthy Status	Study Design, Duration	Markers	References
Hypertension						
Vitamin C, Vitamin E *Vs* Placebo	2 g vitamin C and 800IU vitamin E		39 subjects Hypertension	Double-blind, placebo-controlled study	Lipid hydroperoxides ↓ Methionine-induced homocysteinemia EDD ↔	[175]
Vitamin C, Vitamin E *Vs* Placebo	1 g/day Vitamin C + 400 IU/day Vitamin E		120 subjects Hypertension	Double-blind, randomized, placebo-controlled study 8 weeks	8-iso-PGF_2α_ ↓ Antioxidant capacity ↑ (Na, K)-ATPase ↑	[176]
(-)-epicatechin	100 mg/day		37 Pre-hypertension	Randomized, controlled study 4 weeks	MGO ↔ Advanced glycation end products ↔	[177]
Quercetin 3-glucoside	160 mg/day		37 Pre-hypertension	Randomized, controlled study 4 weeks	MGO ↓ Advanced glycation end products ↔	[177]
Epicatechin *Vs* Placebo	100 mg epicatechin		60 (26 men and 34 female) Borderline or mild hypertension	Repeated-dose, double-blind, placebo-controlled, crossover study 4 weeks	FMD ↑	[178]
Coenzyme Q10 *Vs* Placebo	100 mg/day		60 Mildly hypertension	Randomized, double-blind, placebo-controlled clinical trial	Adiponectin ↑ hs-CRP ↓ IL6 ↓	[179]
Coenzyme Q10 *Vs* Placebo	100 mg twice daily		30 Hypertension and metabolic syndrome	Randomized, double-blind, placebo-controlled crossover trial 12 weeks	Blood pressure ↔	[180]
Diabetes						
Vitamin C and vitamin E	100 IU of vitamin E, 250 mg of vitamin C or 200 IU of vitamin E, 500 mg of vitamin C or 300 IU of vitamin E, 750 mg of vitamin C		9 T1DM children and adolescents	Open-label antioxidant supplementation 6 weeks	TAOC ↔ Endothelial function ↔	[181]
Vitamin C and vitamin E *Vs* Placebo	500 mg of vitamin C twice daily 400 mg of vitamin E twice daily 500 mg Vitamin C plus 400 mg Vitamin E		40 T2DM	Single-blinded randomized controlled clinical trial 90 days	Fasting blood sugar ↓ Lipid profile ↓ HOMA-IR ↓ GSH ↑ MDA ↔	[182]
Vitamin E *Vs* Placebo	804 mg/day		20 subjects with IGT or T2DM	Randomized placebo-controlled crossover design 4 weeks	Oxyphytosterol ↔ oxycholesterol ↔ MDA↔ GSH/GSSG ↔ uric acid ↔	[183]
Vitamin E *Vs* Placebo	400 IU/day		83 T2DM	Double-blind, randomized, controlled clinical trial	Fasting blood sugar ↓ Insulin ↓ Insulin resistance ↓ Paraoxonase-1 activity ↑ TAS ↑ MDA ↔ NO_x_ ↔	[184]
Vitamin C *Vs* GLP1	30 mg/min infusion *Vs* 0.4 pmol/kg/min		20 T2DM	Randomized study	8-iso-PGF_2α_ ↓	[185]
Resveratrol *Vs* Placebo	500 mg/day		60 T2DM patients	Two-arm randomized, double-blind, placebo-controlled clinical trial 3 months	NO ↑ SOD ↑ GSH-Px ↑ CAT ↑ MDA ↓ Insulin ↓ HOMA-IR ↓	[186]
Resveratrol *Vs* Placebo	400 mg/ twice daily		48 T2DM patients	Randomized, placebo-controlled, double-blind clinical trial	MDA ↔ DCFH-DA ↓ FRAP ↑ HOMA-IR ↓	[187]
Resveratrol *Vs* Placebo	500 mg/day or 40 mg/day		192 T2DM patients	Randomized trial 6 months	TAS ↑	[188]
Resveratrol *Vs* Placebo	100 mg/day		50 T2DM patients	Double-blind, randomized, placebo-controlled study 12 weeks	d-ROMs ↓ Cardio-ankle vascular index ↓	[189]
Resveratrol *Vs* Placebo	5 mg twice daily		19 T2DM patients	Randomized, placebo-controlled, double-blind clinical trial 4 weeks	Ortho-tyrosine level ↓ HOMA-IR ↓	[190]
Beta-carotene *Vs* Control Food	0.05 g three times a day		51 T2DM	Randomized double-blinded placebo-controlled crossover clinical trial 6 weeks	NO ↑ GSH ↑ Insulin ↓ HOMA-IR ↓	[191]
Selenium *Vs* Placebo	200 µg/day		60 Diabetic nephropathy patients	Randomized, double-blind, placebo-controlled clinical trial	MDA ↓ GSH ↔ NO ↔	[192]
Selenium *Vs* Placebo	200 μg		60 T2DM	Randomized, double-blind, placebo-controlled trial 8 weeks	TAC ↑ Insulin ↓ HOMA-IR ↓	[193]
α-lipoic acid *Vs* Placebo	600 mg		105 T2DM	Randomized trial 3 months	SOD ↑ GSH-Px ↑ MDA ↓ Fasting plasma glucose ↓ HbA_1c_↓ FPI ↓ HOMA-IR ↓	[194]
α-lipoic acid *Vs* Placebo	300, 600, 900 or 1200 mg/day		38 T2DM	Randomized double-blind placebo-controlled clinical trial 6 months	8-iso-PGF_2α_ ↔ 8-OHdG ↔ Glucose ↓ HbA1c ↓	[195]
R-lipoic acid *Vs* Placebo	R-lipoic acid 600 mg/day		20 subjects with IGT or T2DM	Randomized placebo-controlled crossover design 4 weeks	Oxyphytosterol ↔ oxycholesterol ↔ MDA↔ GSH/GSSG ↔ uric acid ↔	[183]
α-lipoic acid *Vs* Vitamin C	600 mg injection *Vs* 3.0 g Vitamin C injection		90 T2DM	Randomized study 3 weeks	SOD ↑ GSH-Px ↑ MDA ↓ Glucose ↓ HOMA-IA ↓	[196]
Hypercolesterolemia						
N-3 fatty acids	2.4 g/day of a mixture of EPA and DHA		23 women HC	Randomized, controlled, cross over study 6 weeks	SOD ↓ CAT ↑ Cholesterol ↓	[197]
N-3 fatty acids	1.9 g/day of a mixture of EPA and DHA		32 subjects HC	Sequential self-controlled trial 23 weeks	Endothelial function ↔ Platelet function ↓ STAT-8-Isoprostane ↔	[198]
Resveratrol *Vs* Placebo	150 mg/day		18 subjects HC	Randomized study 4 weeks	TAC ↔ Vitamin E ↑ Total cholesterol ↔	[199]
Obesity						
Vitamin C, Vitamin E, Selenium *Vs* Placebo	500 mg of Vitamin C, 400 IU of Vitamin E and 50 μg of selenium 7 d/wk		44 children and adolescents Overweight or obese	Randomized, placebo-controlled, single-masked intervention	8-iso-PGF_2α_ ↓ MDA ↓ Antioxidant status ↑	[200]
Vitamin C	3 g		14 men Overweight/obese grade I	5 min	Protein carbonylation TBARS SOD	[201]
N-6 fatty acids	1000 mg conjugated linoleic acid supplementation 400 IU vitamin E		38 patients Obeses NAFLD	Randomized, controlled clinical trial 8 weeks	MDA ↓ Insulin ↓ HbA1c ↓	[202]
Epigallocatechin-gallate and resveratrol *Vs* Placebo	282 mg/d EGCG, 80 mg/d resveratrol		25 (15 male and 10 women) Overweight and obese	Randomized, placebo-controlled study	Oxidative stress ↓ Inflammation ↓ Adipogenesis ↓	[203]
Resveratrol	1 capsule/day		32 Obeses	Randomized Controlled Trial 28 days	Redox-related genes modulation	[204]
Resveratrol	75 mg		28 Obeses	Double-blind crossover supplementation trial 6 weeks	FMD ↑	[205]
Resveratrol	30, 90 and 270 mg		19 (14 male and 5 female) Overweight/obese	Double-blind, randomized crossover study 1 h	FMD ↑	[206]
α-lipoic acid (+ exercise training)	1.0 g per day		24 (12 male and 12 female) Obeses	Randomized controlled trial 12 weeks	TOS ↓ TAC ↑ Ox-LDL ↓	[207]
α-lipoic acid	1800 mg/day		8 male Overweight and obese	Randomized Controlled Trial 2 weeks	Insulin resistance ↔	[208]
α-lipoic acid	600 mg intravenously once daily		13 obese subjects with IGT (obese-IGT)	Randomized study 2 weeks	ox-LDL-Chol ↓ MDA ↓ 8-iso-PGF_2α_ ↓	[209]
α-lipoic acid, carnosine, and thiamine *Vs* Placebo	7 mg ALA/kg body weight, 6 mg carnosine/kg body weight, and 1 mg thiamine/kg body weight		82 subjects Obeses type 2 diabetic	Randomized double-blind placebo-controlled trial 8 weeks	Glucose ↓ HbA_1c_ ↓ HOMA-IR ↑ Hydroperoxide ↓	[210]
Smoke						
γ-Tocopherol-rich supplementation *Vs* Placebo	500 mg/day	16 healthy subjects Smokers		Randomized, double-blind, placebo-controlled study 7 days	FMD ↑ MPO ↓	[211]
γ-Tocopherol-rich supplementation *Vs* Placebo	500 mg/day	25 healthy subjects Healthy male and female cigarette smokers (more than 10 cigarettes/day; more than 1 year)		Randomized, double-blind, placebo-controlled study 24 h	FMD ↑ 8-iso-15(S)-PGF2a ↓	[212]
Vitamin E *Vs* Placebo	400 IU/day	312 male healthy subjects Current smokers		Randomized placebo-controlled trial 36 months	8-iso-PGF2α ↓	[213]
Fish-oil-derived omega-3 fatty acid supplements	80 mg of eicosapentenoic acid and 120 mg of docosahexanoic acid	54 Healthy subjects Heavy-smoker males (smoking ⩾20 cigarettes per day)		Double-blind, randomized clinical trial 3 months	TAS ↑ TOS ↓	[214]
Resveratrol	500 mg resveratrol/day,	25 healthy subjects Smokers		Randomized, double-blind, crossover trial 30 days	TAS ↑ Triglicerides ↓	[215]
Selenium *Vs* Placebo	200 µg/day	312 male Current smokers		Randomized placebo-controlled trial 36 months	8-iso-PGF2α ↔	[213]

8-iso-PGF_2α_ = 8-iso-prostaglandin F_2α_; AGEs = Advanced glycation end products; CAT = catalase; CRP = c-reactive protein; d-ROMs = diacron reactive oxygen metabolites; DCFH-DA = 2′, 7′-Dichlorofluorescin Diacetate; EDD = endothelium-dependent dilation; FMD = brakial flow-mediated dilation; FPI = fasting plasma insulin; FRAP = Ferric Reducing Antioxidant Power; GSH = glutathione; GSH-Px = glutathione peroxidase; GSSG = oxidized glutathione; HbA_1c_ = Glycated hemoglobin; HC = Hypercholesterolemic; HOMA-IR = Homeostatic Model Assessment for Insulin Resistance; MDA = Malondialdehyde; MGO = Methylglyoxal; NO = Nitric oxide; SOD= superoxide dismutase; TAC = Total Antioxidant Capacity; TAOC = Total plasma antioxidant capacity; TAS = Total Antioxidant Status; TBARS = Thiobarbituric acid reactive substances; TOS = total oxidant status.

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
