# Peer review of "The Role of Antioxidants Supplementation in Clinical Practice: Focus on Cardiovascular Risk Factors"

_antioxidants, 2021, doi:10.3390/antiox10020146_

Round 1

Reviewer 1 Report

This is a very comprehensive review. Just one minor comment: figure 2 has got some misformatting issues. so please correct.

Author Response

Reviewer 1

is a very comprehensive review. Just one minor comment:

1) figure 2 has got some misformatting issues. so please correct.

ANSWER: Amended as suggested

Reviewer 2 Report

Antioxidants; antioxidants-1027282

Title: The role of antioxidants supplementation in clinical practice: focus on cardiovascular risk factors

Cammisotto and co-authors prepared a literature review on clinical markers of reactive oxygen species per chronic cardiovascular diseases and the treatment thereof using various antioxidant supplements. Although having shown promise throughout numerous basic science studies and experimental animal models, the clinical value of antioxidant treatments remains a controversy as the authors have stated in the manuscript. I have one general comment for the manuscript as indicated below.

General: In my opinion, the best parts of the manuscript are the tables (1: direct and indirect methods to evaluate biomarkers of oxidative stress; 2: biomarkers of oxidative stress in patients with cardiovascular risk factor; 3: main characteristics and main results of supplementation studies with antioxidants in cardiovascular risk factors). It also admirable that the authors clearly compile the literature in light of the individual diseases as hypertension, obesity, diabetes, hypercholesterolemia, and the behavior of smoking. However, the premise and the conclusions of the manuscript are generic and, in such manner, it is difficult to see how the manuscript significantly adds new information to the literature. It is already well known clinically that antioxidant treatments are controversial depending on dose, time, and the nature of the cardiovascular disease being treated. Thus, the manuscript would benefit from a more precise narration as to what key questions and/or hypotheses remain unresolved and how the current review endeavors to address such knowledge gaps. Ideally, a review should not only list findings of the literature but also convey distinguished perspectives based on the expertise of the authoring group as to how to best move their particular field of study forward.  

Author Response

Reviewer 2

Cammisotto and co-authors prepared a literature review on clinical markers of reactive oxygen species per chronic cardiovascular diseases and the treatment thereof using various antioxidant supplements. Although having shown promise throughout numerous basic science studies and experimental animal models, the clinical value of antioxidant treatments remains a controversy as the authors have stated in the manuscript. I have one general comment for the manuscript as indicated below.

General: In my opinion, the best parts of the manuscript are the tables (1: direct and indirect methods to evaluate biomarkers of oxidative stress; 2: biomarkers of oxidative stress in patients with cardiovascular risk factor; 3: main characteristics and main results of supplementation studies with antioxidants in cardiovascular risk factors). It also admirable that the authors clearly compile the literature in light of the individual diseases as hypertension, obesity, diabetes, hypercholesterolemia, and the behavior of smoking. However, the premise and the conclusions of the manuscript are generic and, in such manner, it is difficult to see how the manuscript significantly adds new information to the literature. It is already well known clinically that antioxidant treatments are controversial depending on dose, time, and the nature of the cardiovascular disease being treated. Thus, the manuscript would benefit from a more precise narration as to what key questions and/or hypotheses remain unresolved and how the current review endeavors to address such knowledge gaps. Ideally, a review should not only list findings of the literature but also convey distinguished perspectives based on the expertise of the authoring group as to how to best move their particular field of study forward.  

ANSWER: Thanks to the reviewer for the positive comments. In accordance with his/her suggestion we have added more of our experience in the review (See page 6, lines 204-217).

Reviewer 3 Report

The authors describe in their review the clinical supplementation with antioxidants and their potential role for protecting aginst CV risk factors.  They give a good overview of the different compounds and of the potential biomarkers.  Resveratrol should be discussed in the paragraph antioxidants supplementation in clinical practice.

They include in their review numerous small trials, all with intermediate outcomes.  However, there exists a large number of big outcome trials with e.g n-3 fatty acids and EPA administration. In 2019 a landmark trial about the role op EPA adminstration in secondary prevantion was published in the NEJM (Reduce-it trial, Bhatt et al).  The authors state that they reviewed randomized clinical trials where the effect of micronutrient supplementation is examined, yet the most important trials are not discussed.

Author Response

Reviewer 3

The authors describe in their review the clinical supplementation with antioxidants and their potential role for protecting against CV risk factors.  They give a good overview of the different compounds and of the potential biomarkers. 

1) Resveratrol should be discussed in the paragraph antioxidants supplementation in clinical practice.

ANSWER: According to your suggestion, we added the role of resveratrol in clinical practice (See page 8 lines 306-320 lines).

2) They include in their review numerous small trials, all with intermediate outcomes.  However, there exists a large number of big outcome trials with e.g n-3 fatty acids and EPA administration. In 2019 a landmark trial about the role op EPA adminstration in secondary prevantion was published in the NEJM (Reduce-it trial, Bhatt et al).  The authors state that they reviewed randomized clinical trials where the effect of micronutrient supplementation is examined, yet the most important trials are not discussed.

ANSWER: According to your suggestion, we now added randomized clinical trials about the effect of micronutrient supplementation on the cardiovascular outcome (See page 7 lines 245-253 and 261-271; page 8 lines 286-303, page 9 lines 351-358; page 10 lines 377-384).

Reviewer 4 Report

The manuscript entitled “The role of antioxidants supplementation in clinical 2 practice: focus on cardiovascular risk factors” by Cammisotto et al. is well defined for the current research progress on antioxidants supplementation on cardiovascular risk factors. However, with this structure, it is difficult for the reader to understand the points of this paper.

  • Authors need to add a few more keywords
  • It is necessary to add an "introduction part" to the manuscript for the purpose of clarifying the overall feature and status of this review article.
  • Tables 2 and 3 are too extensible for the reader to easily understand the issue.
  • The quality of Fig.1 and Fig.2 should be improved.

Introduction

  1. Biomarkers of oxidative stress and antioxidants supplementation in clinical practice

1.1.

1.2.

1.3.

1.x.

  1. Biomarkers of oxidative stress and antioxidants supplementation in patients with cardiovascular risk factors.

2.1.

2.2.

2.3.

2.x.

Conclusion

Author Response

Reviewer 4

The manuscript entitled “The role of antioxidants supplementation in clinical 2 practice: focus on cardiovascular risk factors” by Cammisotto et al. is well defined for the current research progress on antioxidants supplementation on cardiovascular risk factors. However, with this structure, it is difficult for the reader to understand the points of this paper.

1) Authors need to add a few more keywords

ANSWER: Amended as suggested

2) It is necessary to add an "introduction part" to the manuscript for the purpose of clarifying the overall feature and status of this review article.

ANSWER: As suggested, we have added an introduction to describe the purpose of this review article (see page 3, lines 71-90)

3) Tables 2 and 3 are too extensible for the reader to easily understand the issue.

ANSWER: We know that Tables 2 and 3 could be too extensible for the readers. However, according to the other reviewers, all parameters described are necessary.

4) The quality of Fig.1 and Fig.2 should be improved.

ANSWER: Amended as suggested

5) Introduction

  1. Biomarkers of oxidative stress and antioxidants supplementation in clinical practice
  1. Biomarkers of oxidative stress and antioxidants supplementation in patients with cardiovascular risk factors.

Conclusion

ANSWER: We thank the reviewer for this suggestion. However, we believe that the original structure given to the review allows for better reading and understanding of the manuscript.

Reviewer 5 Report

This manuscript is well written and it is a comprehensive review of the scientific literature in relation to the role of antioxidants to prevent cardiovascular disease.

Review structure is clear and logical. Literature review is up-to-date and relevant. Figures and tables are well presented.

Here some comments to improve the quality of this review:

  1. While the negative effects of antioxidants at higher doses are mentioned, this should be expanded. It is important to clarify which antioxidants should not be 'abused' or over-dosed because of negative side effects.
  2. The difficulties and challenges for measuring ROS directly should be described, for examples their short half lives and lower concentrations in tissues, among others.
  3. Authors should consider including, mentioning, and describing some meta-analysis for clinical trials evaluating some antioxidants for cardiovascular disease prevention.
  4. Figures have formatting errors such as question marks. Please revise.

Author Response

Reviewer 5

This manuscript is well written and it is a comprehensive review of the scientific literature in relation to the role of antioxidants to prevent cardiovascular disease.

Review structure is clear and logical. Literature review is up-to-date and relevant. Figures and tables are well presented.

Here some comments to improve the quality of this review:

  1. While the negative effects of antioxidants at higher doses are mentioned, this should be expanded. It is important to clarify which antioxidants should not be 'abused' or over-dosed because of negative side effects.

ANSWER: As suggested, we have now clarified the negative side effect associated with over-dosed of each antioxidant (see page 7, lines 242-245 and 257-261; page 8, lines 281-285 and 314-320; page 9, lines 327-33o and lines 337-342 and 347-350; page 10, lines 371-373)

  1. The difficulties and challenges for measuring ROS directly should be described, for examples their short half lives and lower concentrations in tissues, among others.

ANSWER: As suggested, we better described these aspects (see page 4, lines 114-122)

  1. Authors should consider including, mentioning, and describing some meta-analysis for clinical trials evaluating some antioxidants for cardiovascular disease prevention.

ANSWER: According to your suggestion, we now added randomized clinical trials about the effect of micronutrient supplementation on the cardiovascular outcome (See page 7 lines 245-253 and 261-271; page 8 lines 286-303, page 9 lines 351-358; page 10 lines 377-384).

  1. Figures have formatting errors such as question marks. Please revise.

ANSWER: Amended as suggested

Round 2

Reviewer 2 Report

Final comment: Ensure that newly added sections in the revised manuscript have the appropriate citation(s).

Reviewer 4 Report

It looks better than before. Thank you for your effort.